# Factors associated with hypertensive disorders of pregnancy in sub-Saharan Africa: A systematic and meta-analysis

**Maereg Wagnew Meazaw**[1,2]*, **Catherine Chojenta**[2], **Muluken Dessalegn Muluneh**[3,4], **Deborah Loxton**[2]

**1** Federal Ministry of Health, Addis Ababa, Ethiopia, **2** Research Centre for Generational Health and Ageing, School of Medicine and Public Health, Faculty of Health and Medicine, University of Newcastle, Newcastle, Australia, **3** School of Nursing and Midwifery, Western Sydney University, Parramatta, Australia, **4** Amref Health Africa in Ethiopia, Addis Ababa, Ethiopia

* maeregwagnewmeazaw@uon.edu.au

**Data Availability Statement:** All relevant data are within the paper and its Supporting Information files.

## Abstract

### Background

Hypertensive disorders of pregnancy (HDP) are common complications of pregnancy globally, including sub-Saharan African (SSA) countries. Although it has a high burden of maternal and neonatal mortality and morbidity, evidence on the risk of the problem is limited. Therefore, the aim of this review was to systematically examine factors associated with HDP among women in SSA countries.

### Methods

Preferred Reporting Items for Systematic Reviews and Meta-Analyses (PRISMA) was followed. Articles conducted in SSA and published in English from January 2000 to May 2020 from electronic databases including MEDLINE, EMBASE, PubMed, and CINAHL were included. Articles, which focused on HDP and found to be relevant through the reference check, were included. Additional articles found through a hand search of reference lists were also included. The quality of papers was appraised using the Critical Appraisal Skills Programme (CASP) scale. Two reviewers independently screened, extracted, and assessed the quality of the articles. STATA 16 software was used to compute the pooled estimated odds ratios for each of the identified associated factor. Both random and fixed effect models were used for analysis. Heterogeneity of the studies and small study bias were checked by $I^2$ and asymmetric test, respectively.

### Results

Twenty-seven studies met the inclusion criteria and included in the systematic review and meta-analysis. Significant associations with HDP were identified through meta-analysis for the following variables: being primiparous (OR: 1.78; 95% CI: 1.11, 2.44), having previous HDP (OR: 3.75; 95% CI: 2.05, 5.45), family history of HDP (OR: 2.73; 95% CI: 1.85, 3.6), and lower maternal educational level (OR: 1.65; 95% CI: 1.17, 2.13). Due to the limited

**Funding:** The authors received no specific funding for this work.

**Competing interests:** The authors have declared that no competing interests exist.

number of studies found specific to each variable, there was inconclusive evidence for a relationship with a number of factors, such as maternal nutrition, antenatal care visits, birth spacing, multiple birth, physical activity during pregnancy, use of contraceptives, place of residency, family size, and other related associated factors.

## Conclusions

The risk of developing HDP is worse among women who have a history of HDP (either themselves or their family), are primiparous, or have a lower maternal educational level. Therefore, investment in women's health needs considered to reduce the problem, and health service providers need to give due attention to women with at increased risk to HDP. Additionally, interventions need to focus on increasing women's access to education and their awareness of potential associated factors for HDP.

## Introduction

Worldwide, hypertensive disorders of pregnancy (HDP) remain among the top causes of maternal and foetal morbidity and mortality. This contributes to the death of a pregnant woman every three minutes and more than nine million deaths every year [1]. HDP cover a spectrum of conditions associated with hypertension during pregnancy and can result in many maternal and neonatal complications, including death [2]. Different international organizations and professional societies have different classifications of HDP [3–5]. For instance, According to the American College of Obstetrics and Gynaecology (ACOG) classification, HDP is a group of diseases and conditions that occur during pregnancy, including preeclampsia and eclampsia, gestational hypertension, chronic hypertension, and preeclampsia superimposed on chronic hypertension [5]. Whereas, according to the International Society for the Study of Hypertension in Pregnancy (ISSHP) classification, HDP classified as chronic hypertension, gestational hypertension, preeclampsia–de novo or superimposed on chronic hypertension and White coat hypertension [4].

The incidence of HDP varies from country to country, and it is estimated that it affects between 2% and 10% of pregnancies every year [6]. Preeclampsia, eclampsia and gestational hypertension remain among the top HDP that cause maternal and foetal morbidity and mortality [6]. A systematic review and meta-analysis conducted on the prevalence of different types of HDP among African mothers showed that the prevalence ranges from 9.2% for superimposed preeclampsia to 49.8% for gestational hypertension [7].

Several studies that were conducted in sub-Saharan Africa (SSA) reported pregnant women experience suffering due to hypertension and it is reported as among the top five leading causes of morbidity and mortality of women and babies [8–10]. Because of this and other factors, SSA countries experience high maternal and newborn mortality [11].

Many researchers have studied the underlying causes of HDP in various settings. However, the exact aetiology of the condition is unknown [12]. Various studies hypothesized the causes of HDP, including preeclampsia/eclampsia, could be problems of placental implantation and the level of trophoblastic invasion [13].

Different studies conducted in different parts of the world showed a range of associated factors for HDP, although are inconclusive as they reported variations among populations and ethno-geographic groups [14]. In addition, there are inconsistencies in reporting determinant

factors in the literature, even for a particular associated factor such as maternal age [15, 16]. Moreover, most of the evidence reported about associated factors of HDP were from developed countries. The studies conducted on associated factors for HDP across SSA countries were at the health facility level with small sample sizes [17].

Overall, to date there have been no systematic reviews or pooled evidence that have comprehensively investigated associated factors for HDP in SSA countries. This study, therefore, the aim of this study was to identify factors associated with HDP in SSA countries.

## Method

### Search strategy and data extraction

MEDLINE, EMBASE, Maternity and Infant Care, and CINAHL databases were searched for articles published from January 2000 to May 2020. In addition, the reference section from identified articles and articles found by hand searching were also reviewed for additional articles. These dates were chosen because many maternal health programs were introduced at that time because of the Millennium Development Goals. Prior to starting this systematic review, we ensured that the research question had not appeared in any existing review using the Cochrane Health Services Research Projects in Progress (HSRProj) and Prospero International Prospective Register of Systematic Reviews (PROSPERO) database registries. Librarians were also consulted during the design of our search strategy and the search for articles from the above databases. Searching terms used for each databases attached in Supplementary Appendices (see S1 Table).

Two independent reviewers (MW, MD) reviewed the titles, abstracts and keywords of every article retrieved by the search strategy according to the eligibility criteria. Subsequently, the articles were screened and independently selected for inclusion in the systematic review and meta-analysis. Full text of the articles were retrieved for further assessment if the information given suggested that the study met the selection criteria or if there was any doubt regarding eligibility of the article based on the information given in the title and abstract.

The quality of the included studies was assessed by two independent reviewers (MW, MD) using a Critical Appraisal Skills Programme (CASP) checklist for quantitative studies [18]. Any disagreement between the reviewers was resolved by discussion. The following criteria were used to assess the quality of the studies: clearly focused research question/objectives, appropriate method, clearly specified target population, adequate sampling techniques and sample size, valid measurement tool, minimal selection bias, appropriate significance level and confidence interval, and relevance of the findings and applicability to our study. Finally, the quality of each paper was rated on a scale from 0 (none of the quality measures met) to 10 (all quality measures met). The quality of each paper was based on the sum of points awarded. Studies were rated as poor quality (score ≤6), medium quality [19, 20], and high quality (≥9). Data extraction and quality assessment forms are given in Supplementary Appendices (see S2 Table).

### Study selection

Studies were only considered if they were conducted in SSA countries and published between January 2000 and May 2020. We only included studies that were published in English and were peer-reviewed. Although there are different definitions and categorizations for HDP, we included studies regardless of their HDP definition. In addition, we retrieved studies that were only undertaken to identify factors for all types of HDP in their study. In other words, studies that were carried out specific to any types of HDP were excluded. For example, if the study only focused on preeclampsia or eclampsia, we excluded it from our review. Finally, we

excluded articles published before January 2000, governmental and non-governmental reports, letters to the editor, other opinion articles, and studies that did not involve women with HDP.

## Data analysis and synthesis

The extracted information is presented in summary form using tables and in narration. The eligibility criteria for studies to be included in the meta-analysis were studies that calculated odds ratios. In addition, in order to run meta-analysis for specific variable in this review, we set four and above studies per variable as eligibility criteria. For associated factors that met the meta-analysis eligibility criteria, a random effects model was utilised to pool the effect sizes of the individual associated factors, taking into account between-study heterogeneity. Because of the low number of cohort and experimental studies (N = 3) for associated factors, we did not include the Relative Risk (RR) in the meta-analysis. The $I^2$ statistic was used to explain the between-study heterogeneity (0–100%), with a higher percentage variation suggesting more heterogeneity or differences among studies. A test of the heterogeneity of each study data set was obtained for the different articles and showed the level of inconsistency ($I^2 > 50\%$), thereby warranting the use of a random effects model in all the meta-analyses. Forest plots were used to present the combined estimate with a 95% CI. Small-study bias was assessed by an asymmetry test through funnel plot test for each variables included in meta-analysis. The Preferred Reporting Items for Systematic Reviews and Meta-Analyses (PRISMA) checklist was employed to present the findings of studies on associated factors for HDP among pregnant women in SSA [21]. Endnote version 9 used to manage search results. All analyses were conducted in STATA 16 software.

## Results

### Description of included studies

After searching the databases, 6849 articles were identified. In addition, 25 articles from hand search articles and 11 articles from the reference section were added for abstract review. After removing duplicate articles, 4805 articles were eligible for title and abstract review. Finally, we included 27 articles in this systematic review and out of 27 studies, 13 studies were included in the meta-analysis study (see Fig 1). The majority of the articles, 15 articles included in this review were cross-sectional [16, 17, 22–35], nine were case-control [15, 36–43], and the remaining three were of a longitudinal study design [34, 44, 45]. Studies from ten countries were included in this review. A relatively larger number of studies were found from Ethiopia, Nigeria and Ghana. Overall, this review included 13,589 pregnant women in SSA. Further studies characteristics and factors assessed are summarized in Table 1.

### Factors associated with HDP

This systematic review and meta-analysis report has been organised for each individual associated factor associated with HDP.

**I. Maternal age.**   Eleven studies found maternal age was an associated factor for HDP [15, 16, 23–25, 28, 30, 33, 36, 39, 40]. In this review, we identified that eight out of ten studies revealed older age women experience a higher odds of developing HDP than younger women [16, 23, 24, 28, 30, 36, 39, 40]. For example, a study conducted in Tanzania found the odds of HDP in pregnant women aged 35 years or older were 5.3 times more as compared to younger women (AOR: 5.32; 95% CI: 2.55, 11.10) [23]. A case-control study conducted in Ethiopia reported that the odds HDP in pregnant women aged 30 years or older were seven times more than younger women (AOR: 6.59; 95% CI: 2.99, 14.50) [36]. Although the age categorization

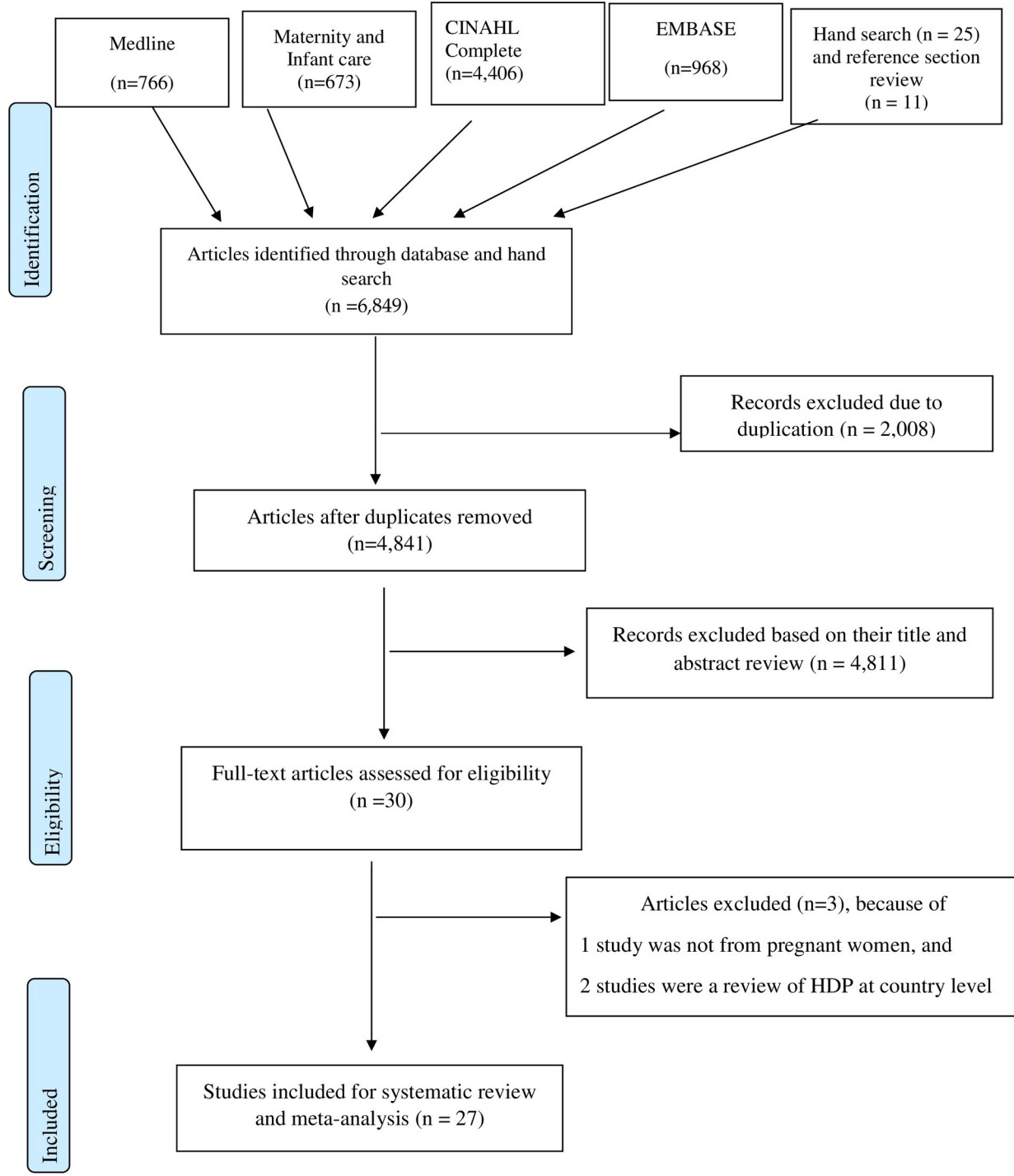

**Fig 1. PRISMA flow chart for selection of studies on factors associated with HDP in SSA.**

differed, similar findings were observed in other studies conducted in Ethiopia, Nigeria, and Ghana [16, 24, 28, 30, 39, 40]. On the other hand, a study conducted in Kenya reported that younger women had a higher odds of developing HDP compared to older women (AOR: 17.0;

**Table 1. Summary table of included studies on associated factors for HDP in SSA countries.**

| Citation | Country | Study design | Study size | Associated factor/s assessed |
|---|---|---|---|---|
| Jennifer Murray et al. [22] | South Africa | Cross-sectional study | 733 | Exposure to sprayed chemical used for malaria control (DDT and DDE) might be associated with elevated risks of HDP in South African women residing in an area sprayed for malaria control. In self-reported HDP was associated with chemicals such as p,p′-DDT (OR:1.50; 95% CI: 1.10, 2.03) and p,p′-DDE (OR:1.58; 95% CI: 1.09, 2.28) in serum concentrations. |
| Tarkie A and Abere W. (2019) [25] | Ethiopia | Cross-sectional study | 422 | Maternal age <24 (AOR: 0.31; 95% CI: 0.05, 3.02), family history of hypertension (AOR: 7.77; 95% CI: 3.037, 19.62), drinking alcohol during current pregnancy (AOR: 1.984; 95% CI: 0.77, 5.108), illiteracy (AOR: 3.30; 95% CI: 0.96, 11.93), low monthly income (AOR: 0.20; 95% CI: 0.05, 1.76), single gestation (AOR: 0.15; 95% CI: 0.02, 1.13), history of chronic hypertension (AOR: 1.82; 95% CI: 0.38, 8.69), and tobacco smoking (AOR: 0.081; 95% CI: 1.47, 25.02) were factors associated with HDP but not statistically significant results. |
| Akwilina W. Mwanri et al. [23] | Tanzania | Cross-sectional study | 910 | For urban areas, mother's age (OR: 1.10; 95% CI: 1.03, 1.20), gestational age (OR: 1.10; 95% CI: 1.02, 1.20), mid-upper arm circumference (MUAC) (OR: 1.13; 95% CI: 1.01, 1.23), dietary diversity score (OR: 1.31; 95% CI: 1.20, 1.60) and being HIV-positive (OR: 2.40; 95% CI: 1.10, 5.18) were independently associated with HDP. In the rural area, the risk of developing HDP increased with the increase in age (OR: 1.12; 95% CI: 1.00, 1.24), gestational age (OR: 1.14; 95% CI: 1.01, 1.30) and marginally significant associated with MUAC (OR: 1.15; 95% CI: 0.99, 1.35). In the rural area, maternal age (AOR:1.12; 95% CI 1.00, 1.24), gestational age in week (AOR: 1.14; 95% CI 1.01, 1.30), and MUAC (AOR: 1.15; 95% CI: 0.99, 1.35) were predictors of HDP |
| Swati Singh et al. [45] | Nigeria | Longitudinal study | 216 | Previous history of preeclampsia (RR: 4.2; 95% CI: 2.14, 6.81), multiple birth (RR: 3.8; 95% CI: 1.04, 6.24), gestational diabetes (RR: 4.8; 95% CI: 1.91, 6.75) and obesity (RR: 2.7; 95% CI: 1.37, 5.51) were factors associated with HDP. |
| Hadiza A. Agbo et al. [17] | Nigeria | Cross-sectional study | 136 | Maternal weight (p<0.009), hip circumference (p<0.018), parity (p<0.043), waist circumference (p<0.001), abdominal height (p<0.040), waist/height (p<0.020), history of developing hypertension in previous pregnancy (p<0.000), birth weight of baby (p<0.02), and caesarean section delivery (p<0.05) were the predictor of HDP |
| Wubanchi Terefe et al. [27] | Ethiopia | Cross-sectional study | 270 | Age of the mother (p<0.67), parity (p<0.95) or antenatal care (ANC) follow up (p<0.096) were not statistically significant factor for HDP. |
| C. T. Ndao et al. [38] | Senegal | Case-control study | 490 | Family history of hypertension (AOR: 2.2; 95% CI: 1.1, 4.5) and presence of placenta malaria infection (AOR: 2.7; 95% CI: 1.0, 7.6) were factors associated with HDP. |
| Tesfaye A. and Tilahun M. (2018) [30] | Ethiopia | Cross-sectional study | 422 | Having family history of HDP (AOR: 5.25; 95% CI: 1.39, 19.86), kidney disease (AOR: 3.32; 95% CI: 1.04, 0.58), asthma (AOR: 37.95; 95% CI: 1.41, 1021) and gestational age ≥37 weeks (AOR: 0.096; 95% CI: 0.04, 0.23) were factors associated with HDP. In addition, family size >3 (AOR: 0.59; 95% CI: 0.13, 2.67), being primiparous (AOR: 1.17; 95% CI: 0.48, 2.86) and MUAC >21 cm (AOR: 0.51; 95% CI: 0.19, 1.35) were also identified as factors but not statistically significant. |
| Pierre Marie Tebeu et al. [41] | Cameroon | Case-control study | 566 | Illiterate mother (AOR: 1.6; 95% CI: 1.0, 2.3), not being in paid employment (OR: 2.8; 95% CI: 1.1, 6.9), nullipara (AOR: 2.8; 95% CI: 1.5, 3.6), family history of hypertension (AOR: 3.6; 95% CI: 1.6, 8.5), history of hypertension during pregnancy (AOR: 7.0; 95% CI: 3.0, 16.4) and history of chronic hypertension (AOR: 2.0; 95% CI 0.5, 7.5) were factors associated with HDP. |
| Hailemariam Berhe Kahsay et al. [42] | Ethiopia | Case-control study | 330 | Rural residence (OR: 3.7; 95% CI: 1.9, 7.1), a low fruit diet (OR: 5.1; 95% CI: 2.4, 11.15), Body Mass Index (BMI) >25 kg/m² (AOR: 5.5; 95% CI: 1.12, 27.6), being diabetes (AOR: 5.4; 95% CI: 1.1, 27.0), being unmarried (AOR: 0.44; 95% CI: 0.12, 1.5), family history of hypertension (AOR: 2.1; 95% CI: 0.7, 6.4), eating vegetable (AOR: 1.2; 95% CI: 0.6, 2.3), no smoking history (AOR: 0.6; 95% CI: 0.07, 5.2), drinking coffee (AOR: 1.9; 95% CI: 0.8, 4.4), multiple birth pregnancies (AOR: 4.2; 95% CI: 1.3, 13.3) and use of oral contraceptive (AOR: 1.2; 95% CI: 0.6, 2.4) were factors associated with HDP. |
| Deborah van Middendorp et al. [31] | Ghana | Cross-sectional study | 1644 | Women in urban settings (p<0.001) was at higher risk of developing HDP compared to rural women. In both rural and urban Ghana, BMI, heart rate, and family history of hypertension were independently associated with raised BP (p<0.001). |
| A. A. Ali et al. (2014) [43] | Sudan | Case-control study | 306 | The highest rate of preeclampsia was in winter (OR: 1.7; 95% CI: 1.1, 2.7) and the lowest rate was in autumn (0.2%) (OR: 0.8; 95% CI: 0.4, 1.8). |

*(Continued)*

**Table 1.** (Continued)

| Citation | Country | Study design | Study size | Associated factor/s assessed |
|---|---|---|---|---|
| Getinet Ayele et al. [36] | Ethiopia | Case-control study | 466 | Age of mothers >30 years (AOR: 6.59; 95% CI: 2.99, 14.50), lack awareness of risk of hypertension (AOR: 8.24; 95% CI: 1.87, 35.96), absence of chronic disease (AOR: 0.14; 95% CI: 0.05, 0.43), primiparity (AOR: 5.09; 95% CI: 1.23, 21.02), frequent salt consumption (AOR: 4.41; 95% CI: 1.25, 15.56), BMI >30 kg/m$^2$ (AOR: 9.91; 95% CI: 4.29, 22.86) and previous HDP (AOR: 2.85; 95% CI: 1.27, 6.39) were factors associated with HDP. |
| Mastewal Arefaynie Temesgen [37] | Ethiopia | Case-control study | 470 | Previous history of preeclampsia (AOR: 4.22; 95% CI: 2.06, 8.65), family history of hypertension (AOR: 3.94; 95% CI: 1.98, 7.83), illiterate women (only read and write) (AOR: 2.64; 95% CI: 1.106, 6.32) were factors associated with HDP. |
| Emmanuel Ratemo Omenya et al. [15] | Kenya | Case-control study | 344 | Use of hormonal contraceptives (AOR: 29.5; 95% CI: 10.3, 94.3), physical inactivity (AOR: 1.63; 95% CI: 0.47, 5.86), being age < 20 years (AOR: 17.0; 95% CI: 5.0, 73.2), abnormal BMI >30 kg/m$^2$ (AOR: 9.90; 95% CI: 4.29, 22.86), smoked cigarettes (AOR: 5.8; 95% CI: 1.6, 25.3), sexual relationship with other partners (AOR: 17.0; 95% CI: 5.0, 73.2), drinking alcohol (AOR: 29.5; 95% CI: 10.3, 94.3), primary education (AOR: 1.5; 95% CI: 0.9, 2.7), being nulliparous (AOR: 1.5; 95% CI: 1.0, 3.5), family history of HDP (AOR: 17; 95% CI: 5.0, 73.2), previous history of HDP (AOR: 22.5; 95% CI: 3.0, 170) were factors associated with HDP. |
| Samuel Azubuike and Ibrahim Danjuma [24] | Nigeria | Cross-sectional study | 159 | Previous history of HDP, BMI, family history of HDP, maternal age, primiparous, history of still birth, history of induced delivery, being diabetic, oral contraceptive use, history of hypertension during pregnancy were factors associated with HDP (p<0.0005). |
| P.N. Ebeigbe et al. [16] | Nigeria | Cross-sectional study | 442 | Being nulliparous (p<0.0001), maternal age >40 years (p<0.0001), absence of ANC visit (p<0.001) were factors associated with HDP. |
| Zenebe Wolde et al. [26] | Ethiopia | Cross-sectional study | 158 | Rural residence (P<0.0001), multiparous women (p<0.511) and absence of ANC follow-up (p<0.166) were factors identified as risk for developing HDP. |
| Liyew Mekonen et al. [28] | Ethiopia | Cross-sectional study | 408 | Maternal education (AOR: 2.5; 95% CI: 1.2, 5.3), maternal age (AOR: 2.73; 95% CI: 1.31, 5.7), previous history of preeclampsia (AOR: 19.3; 95% CI: 5.2, 72.1), family history of preeclampsia (AOR: 7.2, 95% CI: 2.9, 17.8), and Primigravida (AOR: 1.6; 95% CI: 0.8, 2.8) were factors associated with HDP. |
| W.K.B.A. Owiredu et al. [30] | Ghana | Cross-sectional study | 150 | Women between 35–39 years of age (AOR: 9.2; 95% CI: 2.5, 34.7), being obese (AOR: 4.7; 95% CI: 1.7, 12.5), family history of hypertension (AOR: 6.8; 95% CI: 2.3, 19.6), women whose partners used a condom during coitus (AOR: 5.8; 95% CI: 1.2, 23.0), changing partners (AOR: 2.3; 95% CI: 1.1, 5.8), contraceptive use (AOR: 1.7; 95% CI: 1.2, 3.9), and no physical exercise (AOR: 1.3; 95% CI: 0.6, 2.9) were factors associated with HDP. |
| Leta Hinkosa et al. [39] | Ethiopia | Case-control study | 199 | Maternal age ≥35 (AOR: 2.51; 95% CI: 1.08, 5.83), rural residence (AOR: 1.79; 95% CI: 1.15, 2.8), being primiparous (AOR: 3.39; 95% CI: 2.16, 5.33), nullparity (AOR: 4.35; 95% CI: 2.36, 8.03), positive history of abortion (AOR: 4.39; 95% CI: 1.64, 11.76), twin pregnancy (AOR: 3.78; 95% CI: 1.52, 9.39), lack of ANC visits (AOR: 3.05; 95% CI: 1.56, 5.96), previous HDP (AOR: 3.81, 95% CI: 1.69, 8.58), having family history HDP (AOR: 5.04; 95% CI: 2.66, 9.56) and history of diabetes mellitus (DM) (AOR: 5.03; 95% CI: 1.59, 15.89) were factors associated with HDP. |
| Larry Jones et al. [40] | Ghana | Case-control study | 216 | Maternal age >35 (AOR: 3.24; 95% CI: 0.96, 10.94), unemployed (AOR:1.03; 95% CI: 0.31, 3.42), married (AOR: 1.57; 95% CI: 0.73, 3.40), eating fatty food (AOR: 4.42, 95% CI: 2.25, 8.66), family history of HDP (AOR: 4.41; 95% CI: 1.83, 10.58), history of preterm delivery (AOR: 4.66; 95% CI: 1.37, 15.86) were factors associated with HDP. |
| W.K.B.A. Owiredu et al. [32] | Ghana | Cross-sectional study | 1005 | HDP associated with being at extreme age and delivery of low birth weight (p<0.001). In addition, increase incidence of HDP in rainy season. |
| Olivier Pancha Mbouemboue et al. [33] | Cameroon | Cross-sectional study | 160 | Maternal age (P<0.013), previous twin pregnancy (P<0.013), previous preeclampsia (P<0.013) and multiparous (P<0.001) were factors associated with HDP. |
| B Longo-Mbenza et al. [44] | DR Congo | Longitudinal study | 238 | Physical activity during pregnancy (RR: 0.63; 95% CI: 0.33, 0.94) and greater than 3 times daily servings of vegetables (RR: 8.8; 95% CI: 0.6, 0.98) were significantly associated with increased risk of developing HDP (p<0.01). |
| Edward Antwi, et al. (2016) [34] | Ghana | Cohort study | 2,529 | For gestational hypertension, parity (AOR: 0.90; 95% CI: 0.66, 1.23), maternal weight (AOR:1.02; 1.01 to 1.03), maternal height (AOR: 0.97; 95% CI: 0.95, 0.99), diastolic BP (AOR:1.04; 95% CI: 1.03, 1.06), family history of hypertension (AOR:1.46; 95%CI: 1.06, 2.02) and having gestational hypertension in a previous pregnancy (AOR:9.55; 95% CI:5.42, 16.84) were the associated factors. |

(*Continued*)

**Table 1.** (Continued)

| Citation | Country | Study design | Study size | Associated factor/s assessed |
|---|---|---|---|---|
| V. O. Osunkalu, et al. [35] | Nigeria | Cross-sectional | 200 | Mean plasma homocysteine level and malondialdehyde were significantly higher in women with preeclampsia and gestational hypertension when compared to normotensive women (p 0 < .05). However, MTHFR enzyme level, glutathione (GSH), superoxide dismutase (SOD) and catalase (CAT) were significantly higher in normotensive women as compared to women with preeclampsia and gestational hypertension (p < 0.05). Preeclampsia was significantly associated with an increased risk of lipid peroxidation (OR = 4.923; p < 0.007). |

95% CI: 5.0, 73.2) [15]. We did not run meta-analyses for age due to the multiple age categori-sations amongst the studies.

**II. Parity.**   In this review, we identified eleven studies that investigated the relationship between number of pregnancies and developing HDP [15, 16, 23, 24, 28–30, 33, 36, 39, 41]. Evidence on developing HDP with experience of pregnancy was mixed. Five studies found that primiparous women had an increased odds of developing HDP [24, 28, 29, 36, 39], three of which found a significant relationship [24, 36, 39]. In contrast, three other studies revealed that being nulliparous was a significant associated factor for HDP [15, 16, 41]. For example, a study conducted in Cameroon found that women who had never been pregnant were nearly threefold increased odds of developing HDP in comparison with multiparous mothers (OR: 2.8; 95% CI: 1.5, 3.6) [41]. However, a case-control study conducted in Ethiopia found a significant increase in odds of HDP for both primiparous (AOR: 3.39; 95% CI: 2.16, 5.33) and nulliparous women (AOR: 4.35; 95% CI: 2.36, 8.03) as compared to multiparous women [39]. Overall, we included four studies in the meta-analysis that analysed the odds of developing HDP in primiparous women. The other variables (null parity and multiparty) did not meet the criteria to undertake a meta-analysis. Both a funnel plot and Egger's test showed there was no small study bias in the included studies (see S1 Fig in S1 File).

The result revealed there was a positive association between primiparity and HDP. We applied a fixed effect model since heterogeneity was low ($I^2$ <50%). Primiparous women have nearly a twofold increased odds of developing HDP compared to multiparous women (OR: 1.78; 95% CI: 1.11, 2.44) (Fig 2).

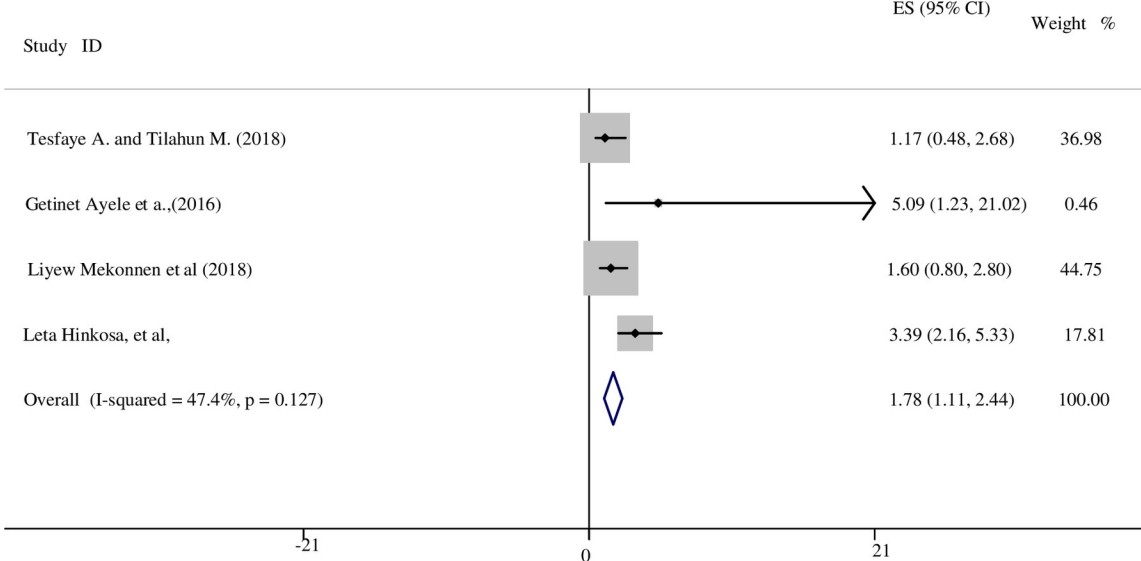

**Fig 2. The association between parity and HDP in SSA.**

**III. Previous HDP.**   In this review, we found ten studies that assessed the odds of developing HDP for mothers with previous HDP [15, 24, 28, 33, 34, 36, 37, 39, 41, 45]. All of these studies found that having previous experience of HDP had a positive association with present HDP. For instance, a longitudinal study conducted in Nigeria reported that having HDP previously increased the likelihood of having HDP in the present pregnancy by four times compared to women without any history of HDP (RR: 4.2; 95% CI: 2.14, 6.81) [45]. Similarly, a study in Cameroon also reported that mothers with a history of hypertension during a previous pregnancy were at a sevenfold increase the odds of developing HDP in their subsequent pregnancies compared to women without history of HDP (AOR: 7.0; 95% CI 3.0, 16.4) [41].

Six studies met the meta-analysis criteria and the final finding revealed a significant association between having HDP in previous pregnancies and developing HDP in the current pregnancy. Based on pooled estimates of OR, the odds of women who had a history of HDP were nearly four times more likely to develop HDP compared to women who had no history of HDP (OR: 3.75; 95% CI: 2.05, 5.45). Heterogeneity was checked and a random effects model was used in this analysis (Fig 3). Both a funnel plot and Egger's test showed there was no small study bias in the included studies (see S2 Fig in S1 File).

**IV. Family history of HDP.**   For this variable, we found fourteen studies that had analysed the relationship between having a family history of HDP and the developing HDP [15, 17, 24, 27–31, 34, 37–41]. All of these findings showed there was a positive association between family history of HDP and HDP in pregnant women. A study conducted in Ethiopia reported that the odds of women with a family history of preeclampsia were six times more likely to develop HDP compared to women without a family history of preeclampsia (AOR: 6.2; 95% CI: 2.9, 12.8) [28]. Another study from Ghana also reported women with family history of hypertension during pregnancy have seven times higher odds of developing HDP (AOR:6.8; 95% CI: 2.3, 19.6) [30].

Overall, ten studies fulfilled the inclusion criteria and were included in the meta-analysis. Heterogeneity was tested for through a fixed effect model. The finding of the meta-analysis showed that the odds of women who have a family history of HDP were nearly three times

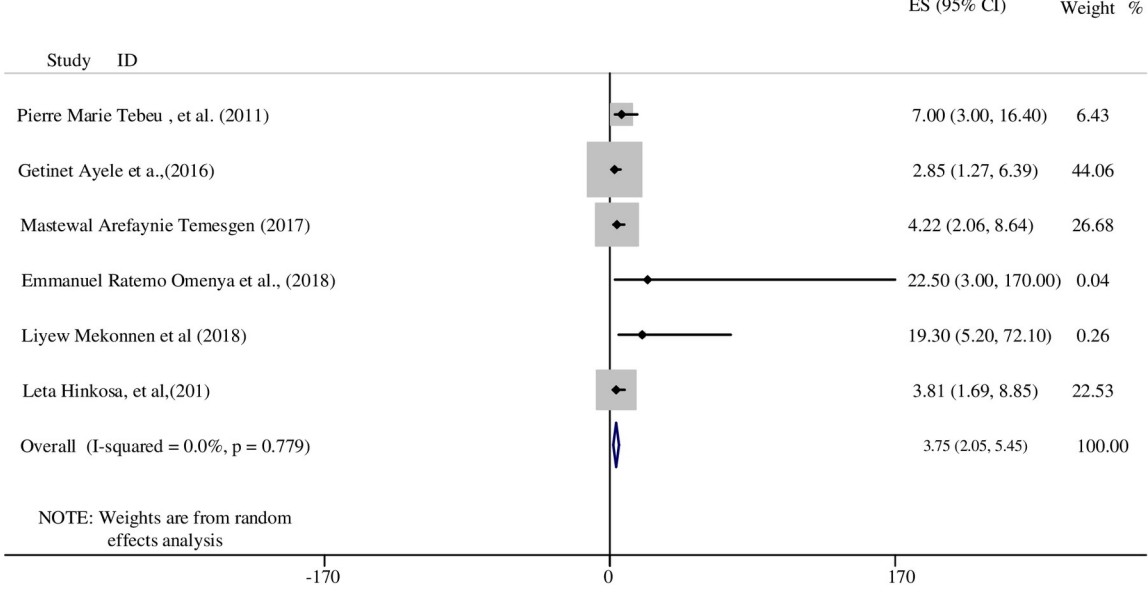

**Fig 3. The association between previous HDP and HDP in SSA.**

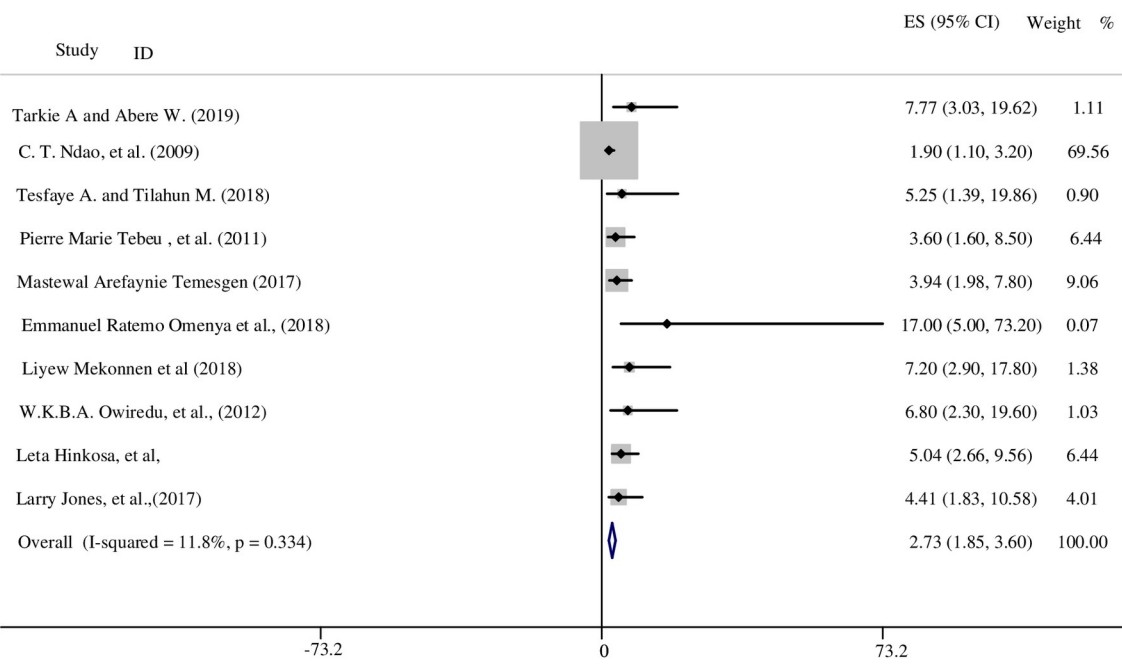

**Fig 4. The association between family history of HDP and HDP in SSA.**

more likely to develop HDP compared with women without a family history of HDP (OR: 2.73; 95% CI:1.85, 3.6) (Fig 4). Both a funnel plot and Egger's test showed there was no small study bias in the included studies (see S3 Fig in S1 File).

**V. Maternal educational status.** In this review, we found seven studies that analysed the association between level of maternal education and HDP [15, 23, 28–30, 37, 41]. All of the research findings revealed that having a low educational attainment was positively associated with HDP in comparison with having a higher level of education. For instance, a study conducted in Ethiopia reported that the odds of women with no education were 2.5 times more likely to develop HDP compared with women with secondary and above education level (AOR: 2.5; 95% CI: 1.2, 5.3) [28]. Similarly, another study conducted in Ethiopia also reported that women who only have a low level of education were 2.6 times more likely to develop HDP compared to women who have a higher or above education level (AOR: 2.64; 95% CI: 1.10, 6.32) [37].

We included all seven studies in the meta-analysis, and the result showed that there was a significant association between women's level of education and HDP. We used a fixed effect model for this variable. The funnel plot test also report no small study bias in the included studies (see S4 Fig in S1 File). Based on pooled estimates of OR, the odds of women who had a lower education level were 1.65 times more likely to develop HDP compared to women who had a higher or above education level (OR: 1.65; 95% CI: 1.17, 2.13) (Fig 5).

**VI. Maternal body mass index and mid-upper arm circumference.** In this review, we found nine studies that reported on the association between a high BMI and HDP [17, 23, 24, 29–31, 36, 42, 45]. Out of those nine studies, seven studies revealed a significant association between high BMI and HDP [23, 24, 30, 31, 36, 42, 45]. There was inconsistency in the categorization of BMI and MUAC measurement across the studies. A cross-sectional study conducted in Tanzania among urban and rural women attending ANC revealed that the odds of women with an upper quantile MUAC ($\geq$29.1 cm) had an more likely to developing HDP as compared to women with a lower quantile MUAC ($\leq$24.5 cm) (AOR: 1.12; 95% CI: 1.1, 1.7)

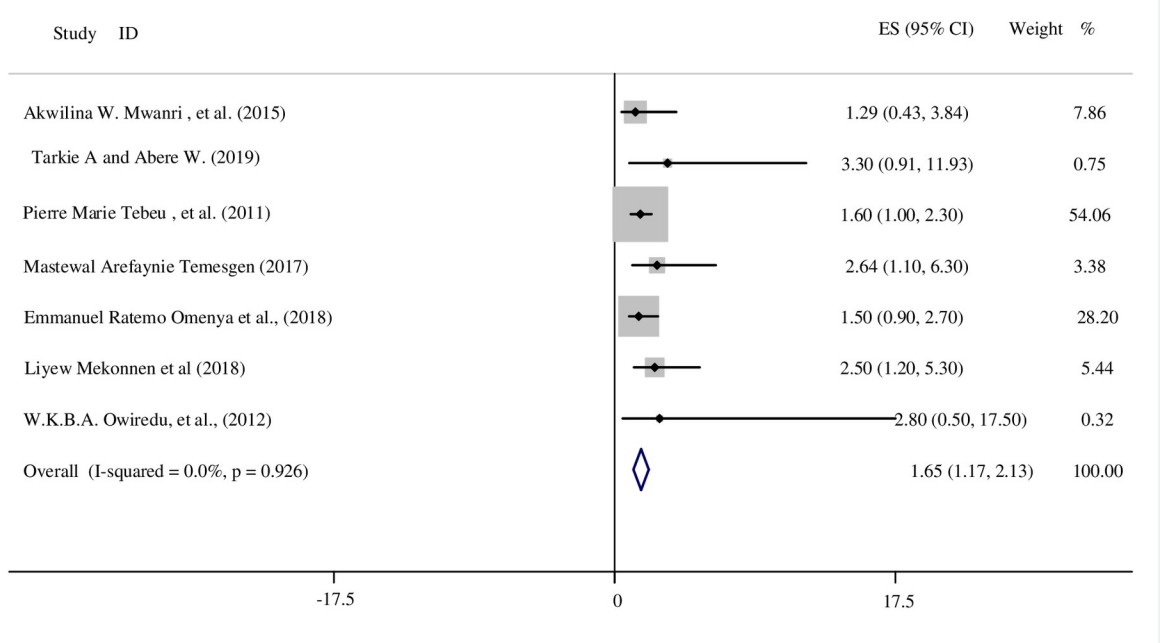

**Fig 5. The association between level of maternal education and developing HDP in SSA.**

[23]. A longitudinal study from Nigeria reported that obese women were nearly three times more likely to develop HDP compared to women with a normal BMI (RR: 2.7; 95% CI: 1.37, 5.51) [45]. Studies conducted in Ethiopia and Ghana showed that overweight (BMI >25 kg/m$^2$; AOR: 5.5; 95% CI: 1.12, 27.6) and obese (BMI>30 kg/m$^2$; AOR: 4.7; 95% CI: 1.7, 12.5) women were nearly five times more likely to develop HDP compared to women with a normal BMI, respectively [30, 42]. Another study conducted in Ghana reported high BMI was an independent associated factor for high blood pressure during pregnancy in both urban and rural pregnant women [31]. For this variable, we did not run a meta-analysis due to the different measurement units (BMI and MUAC) used among the studies.

**VII. Maternal diabetes mellitus.** Five studies investigated the risk of developing HDP in pregnant women with DM [17, 23, 39, 42, 45]. Of these five studies, three studies found maternal DM was positively associated with HDP [39, 42, 45]. A study conducted in Nigeria found that pregnant women with DM were five times more likely to develop HDP in comparison to women without DM (RR: 4.8; 95% CI: 1.91, 6.75) [45]. Similarly, two studies from Ethiopia also reported that the odds of developing HDP was 5.4 (AOR: 5.4; 95% CI: 1.1, 27.0) and 5.03 (AOR: 5.03; 95% CI: 1.59, 15.89) times higher among diabetic mothers compared to non-diabetic mothers [39, 42]. Four studies were met the meta-analysis criteria and included in the meta-analysis. The result revealed there was no association between having DM and HDP (OR: 1.11; 95% CI: 0.30: 1.92) (Fig 6). A random effects model was used for this variable to test for heterogeneity and funnel plot test report no small study bias in the included studies (see S5 Fig in S1 File).

**VIII. Pregnancy related factors.** In this review, we found different pregnancy related associated factors that were associated with developing HDP. We found five studies that reported the number of gestations as associated factor for developing HDP [25, 33, 39, 42, 45]. Out of five studies, four reported a significant association between multiple birth and increased odds of developing HDP [33, 39, 42, 45]. For instance, a study conducted in Ethiopia reported that women with twin pregnancy had almost a fourfold higher odds of developing HDP

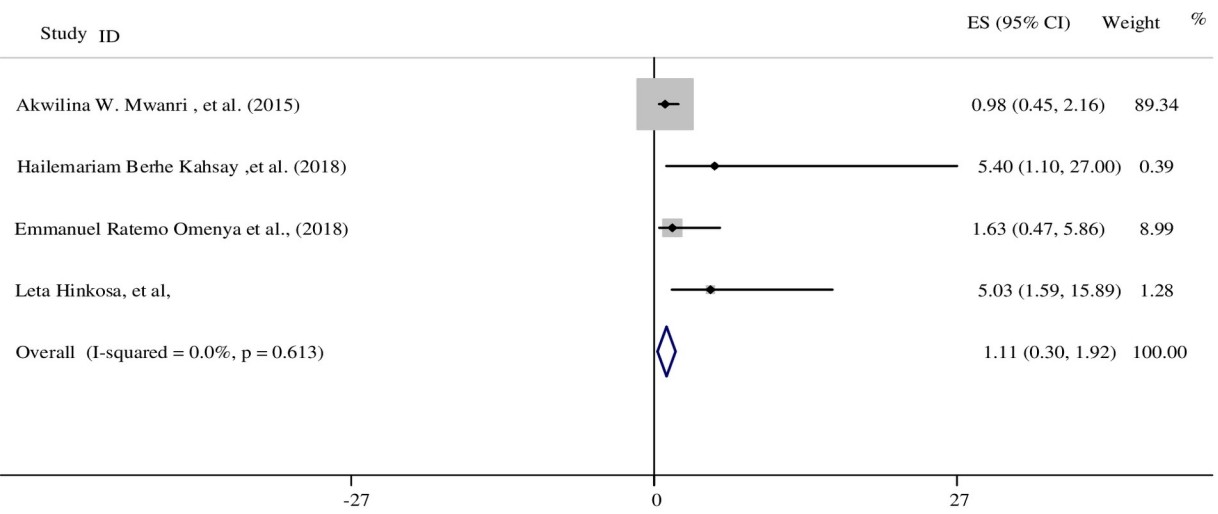

**Fig 6. The association between maternal diabetes mellitus and developing HDP in SSA.**

compared to women with singleton pregnancy (AOR: 3.78, 95% CI: 1.52, 9.39) [39]. This review also found three studies that examined the relation of having abortion history on the development of HDP in subsequent pregnancies. However, only one study, which was conducted in Ethiopia, reported a significant association between the two variables, and found that the odds of women with an induced abortion history were 4.4 times more likely to develop HDP compared to women with no abortion history (AOR: 4.39; 95% CI: 1.64, 11.76) [39]. Another study found that pregnant women with a history of preterm delivery were 4.66 times more likely to develop HDP, and the result was also statistically significant (AOR: 4.66; 95% CI: 1.37, 15.86) [40]. Similar to preterm delivery history, two studies from Nigeria and Tanzania reported that stillbirth history was associated factor for developing HDP [23, 24]. However, only one study conducted in Nigeria reported a significant association between previous stillbirth and HDP (p<0.0001) [24]. Furthermore, two studies showed that pregnant women with no ANC visits were at a higher odds of developing HDP [16, 39]. For example, a study conducted in Ethiopia reported that the odds of developing HDP in women who had no ANC visits were three times more as compared to women who did have ANC visits (AOR: 3.05, 95% CI: 1.56, 5.96) [39].

**IX. Pre-existing medical conditions.** In this review, we found that women with a pre-existing medical condition had increased odds of developing HDP. Two studies reported that women with chronic hypertension were more likely to develop HDP compared to normotensive women, although the results were not statistically significant [25, 41]. Similarly, one study reported that pregnant women with a family history of chronic hypertension were at an increased oddsof developing HDP, although this was not statistically significant either [42]. In addition, based on a study conducted in Ethiopia, pregnant women with kidney disease during pregnancy had more than a three times higher odds of developing HDP compared to pregnant women without kidney disease (AOR: 3.32; 95% CI: 1.04, 10.58) [29]. Furthermore, a cross-sectional study conducted in Tanzania reported that HIV positive women were at a 2.4 times increased risk of developing HDP compared to HIV negative women (AOR: 2.4; 95% CI: 1.10, 5.18) [23]. A study conducted in Ethiopia reported that absence of chronic diseases such as cardiac, renal or diabetic disease experienced 86% less HDP during pregnancy in comparison with those with a chronic disease (AOR: 0.14; 95% CI: 0.05, 0.430) [36].

**X. Nutrition and related factors.** In this review, several diet and nutrition related factors were identified as factors for developing HDP. A study conducted in Ethiopia reported women who consumed low amounts of fruit in their diet had five times higher odds of developing HDP than women who did consume fruit regularly (AOR: 5.1; 95% CI: 2.4, 11.15) [42]. Similarly, a longitudinal study conducted in DR Congo reported that consuming three or more daily servings of vegetables was a significant protective factor against developing HDP compared to less vegetable intake (RR: 0.8; 95% CI: 0.6, 0.98) [44]. Another study conducted in Ethiopia showed that women with an excessive and frequent dietary intake of salt during pregnancy were 4.4 times more likely to develop HDP compared to women with a normal amount of salt in their food (AOR: 4.4; 95% CI: 1.25, 15.54) [36]. In addition to salty food, a study in Ghana reported that consumption of industrial trans fatty foods such as biscuits, pies, cakes, doughnuts and crackers during pregnancy increased odds of developing HDP nearly five times compared to women who did not consume such foods (AOR: 4.42; 95% CI: 2.25, 8.66) [40]. In a study conducted in Tanzania, the researcher studied daily nutrient adequacy and the quality of pregnant women's diets were assessed using the Dietary Diversity Score (DDS) by measuring the total amount of food groups consumed in the past 24 hours. The result showed that the risk of developing HDP was six times higher in high DDS groups compared to low and medium DDS groups (AOR: 5.84; 95% CI: 2.11, 16.15) [23].

**XI. Sociodemographic and behavioural associated factors.** In this review, we found four studies that reported being physically active during pregnancy was protective against developing HDP [15, 23, 30, 44]. Although inconsistency in the categorization of the level of physical activity during pregnancy among studies, only one out of the four studies revealed a statistically significant result, women who were physically active during pregnancy were 37% less likely to develop HDP compared to inactive women (RR: 0.63; 95% CI: 0.33, 0.94) [44]. On the other hand, three studies investigated the association between drinking alcohol during pregnancy and the risk of developing HDP [15, 25, 30]. However, only one study, which was conducted in Kenya, reported a significant association between alcohol drinking during pregnancy and increased odds of developing HDP (AOR: 29.5; 95% CI: 10.3, 94.3) [15]. The same study also reported that smoking during pregnancy increased the odds of developing HDP by nearly six times compared to non-smoking pregnant women (AOR: 5.8; 95% CI: 1.6, 25.3) [15]. Another study from Ethiopia revealed that not smoking during pregnancy was protective against developing HDP, but the finding was not statistically significant (AOR: 0.6; 95% CI: 0.07, 5.2) [42]. This study also reported that coffee drinking during pregnancy doubled the odds of developing HDP compared to women who did not drink coffee during their pregnancy, although the finding was not statistically significant either (AOR: 1.9; 95% CI: 0.8, 4.4) [42].

Further to the above studies finding, three studies analysed the marital status of women and its association with developing HDP [30, 40, 42]. However, none of the findings reported statistically significant results on the relationship between the two variables. On the other hand, a study conducted in Ghana revealed that women who changed their partners in their second or subsequent pregnancies had more than a two times higher odds of developing HDP in comparison with women who were pregnant with the same partner (AOR: 2.3; 95% CI: 1.1, 5.8) [30]. Interestingly, this study also reported that women whose partner used a condom during coitus were at a six times increased odds of developing HDP compared to coitus without a condom (AOR: 5.8; 95% CI: 1.2, 23.0) [30].

A mixed result was reported on the effect of residency on the risk of developing HDP. A study conducted in Tanzania reported that women who live in urban settings have twice odds of developing HDP compared to rural dwellers (AOR: 1.90; 95% CI: 1.1, 3.6) [23]. In contrast, three studies reported that HDP occurs more frequently among women residing in rural

settings [26, 39, 42]. For instance, two studies conducted in Ethiopia indicated that women with a rural residence were nearly two to four times more likely to experience HDP (AOR: 1.79, 95% CI: 1.15, 2.8) in comparison to women with an urban residence (AOR: 3.7; 95% CI: 1.9, 7.1) [39, 42].

Women's employment status was also found associated with HDP. Three studies investigated the employment status of women and the odds of developing HDP [23, 40, 41]. For example, a study conducted in Cameroon revealed that the odds of experiencing HDP were almost three times higher among women who were housewives than women who were employed (AOR: 2.8; 95% CI: 1.1, 6.9) [41]. Furthermore, a study conducted in Ethiopia reported that the odds of women who had no awareness of the risks of hypertension during pregnancy were more likely to develop HDP compared to women who did have this awareness (AOR: 8.24; 95% CI: 1.89, 35.96) [36].

**XII. Other associated factors.** In four studies, researchers found that contraceptive use were was positively associated with experience of HDP during pregnancy [17, 24, 30, 42]. A study conducted in Ghana reported that the occurrence of HDP was found to be higher among women that used a contraceptive compared to women who did not use a contraceptive (AOR: 1.7; 95% CI: 1.2, 3.9) [30]. In other studies, the risk of developing HDP was found to be higher in women who used a hormonal contraceptive [17, 24, 42]. For instance, a study in Nigeria found a significant association between the use of an oral contraceptive and the risk of developing HDP (P<0.0005) [24].

In addition, seasonal variation [32, 43] and exposure to insecticide chemicals used for malaria control [22] were identified as associated factors for developing HDP. However, this systematic review was not found sufficient evidence to draw strong to consider as associated factor for developing HDP.

## Discussion

This systematic review identified that primiparity, history of HDP, family history of any type of HDP, and low maternal education level are significantly associated with HDP among SSA women. However, we did not find a statistical association between DM and HDP. This comprehensive systematic review and meta-analysis provides valuable information on the overall associated factors that contribute to the higher burden of HDP in SSA countries.

Primiparous women are at increased odds of developing HDP. This was consistent with studies conducted in Ethiopia, Colombia and China, which reported primiparity meant women are at a higher risk of developing HDP (preeclampsia or eclampsia) compared with multiparous women [46–48]. This was also supported by a systematic review conducted by Zhong-Cheng Luo et al. whose pooled finding showed there was a higher risk of developing preeclampsia for primiparous women compared to multiparous women (OR: 2.42; 95% CI: 2.16, 2.7) [49]. Similarly, a large prospective cohort study (between 1987 and 2004) of women in Sweden showed that the incidence of preeclampsia was 4.1% in primiparous women and 1.7% in multiparous women [50]. This could be explained by the nature of the immune maladaptation of the disease, which elevates the risk of developing HDP among primiparous women compared to multiparous women [51]. In fact, every woman will be at increased risk of complications in her first pregnancy, this review also support being premiparous found to be associated factor for developing HDP. Therefore, special attention should be given to primipara women during ANC visits.

Women who have experienced previous HDP are more likely to experience the recurrence of HDP in consecutive pregnancies. In this study, we found women who had a history of HDP were nearly four times more likely to develop HDP compared to women who had no history

of HDP (OR: 3.75; 95% CI: 2.05, 5.45). Similar findings were also observed in studies conducted in Asia and the UK [52, 53]. This can be explained by the recurrent nature of the disease. However, in some studies the recurrence of HDP was related to the interval between pregnancies. For example, a systematic review aimed at identifying the effect of inter-pregnancy interval on the risk of recurrent HDP reported that a shorter interval between pregnancies (2–4 years) was not associated with an increased risk of recurrent HDP; however, the risk appeared to increase in longer inter-pregnancy intervals [54]. Furthermore, we found having a family history of any type of HDP increased the chance of developing HDP. Consistent with this finding, studies conducted in China, Ethiopia and Brazil reported significant higher odds of developing HDP (preeclampsia, eclampsia, and gestational hypertension) among women who have a family history of any type of HDP compared to women with no such history [48, 55, 56]. Although this could be strongly correlated with the genetic component in pathophysiological abnormalities of preeclampsia and eclampsia [57–59], the cause behind the genetic aspect of this disease is still not known [60]. These results suggest that during ANC visits a detailed individual and family history of HDP should be taken. This would allow early identification of higher risk women and reduce the complications and serious health outcome to the mother and newborn.

Unlike several studies which reported diabetic women were at an increased risk of developing HDP [56, 61], our review did not find this association. It is possible that we did not obtain similar results because of the small number of studies that were included in this meta-analysis. Even though our study did not show a significant relationship, because of the increasing trend of chronic diseases in SSA pregnant women might be at higher risk of developing HDP. Therefore, countries in SSA should pay more attention to this issue and need a routine screening of DM in pregnant women in order to reduce the negative consequences to the mother and foetus.

In our study, we found a significantly higher risk of developing HDP occurred in women who had a low educational level. In agreement with this, studies conducted in various countries reported low educational level put women at a higher risk of developing HDP [61, 62]. Women in SSA countries are mostly uneducated and may not understand how important early identification of associated factors and early initiation of ANC visits are. Therefore, they are likely to be at a higher risk of developing negative health outcomes.

Although we cannot draw conclusions due to difference in categorization of level of physical activity during pregnancy and identified studies does not fulfil our meta-analysis criteria, this review highlights that doing physical activity during pregnancy was found to be protective against developing HDP. Regular physical activity recommended for pregnant and postpartum women for maternal, foetal, and neonatal wellbeing [63–65]. However, the relationship between HDP and the level of exercise needs further investigation. In addition, a number of studies also reported there was a higher risk of developing HDP among oral contraceptive users. In most SSA countries the contraceptive prevalence rate is increasing [66], and hormonal contraceptives are mostly the choice of many women [67]. Therefore, this also needs further investigation. Furthermore, we have a good understanding of how important it is to consider nutritional and related factors to reduce the risk of developing the disease. While it is important to have regular visits to antenatal clinics, there remains the question of how much detailed information should be taken by the health providers about the pregnant woman's individual and family history of the disease, especially in SSA countries [68]. Therefore, it is highly recommended that healthcare professionals assess each pregnant woman's previous history and family related risks during her booking visit and tailor her ANC services.

## Strengths and limitations

This is the first systematic review to quantitatively summarize and conduct a meta-analysis of associated factors for developing HDP in SSA countries. A rigorous search was conducted from multiple electronic databases, the reference sections of identified papers and hand search. A quality assessment was conducted, and two independent reviewers conducted the screening. Nonetheless, there are limitations to our study. All studies used in the meta-analysis were cross-sectional and case-control studies and hence do not show causality. In addition, this review included all types of HDP in meta-analysis and we did not run subgroup analysis due to the limed number of studies per factor. Therefore, this might not show the association between associated factors and each type of HDP. Only studies published in English were included, and as a result, papers published in other languages would have been missed. Heterogeneity of the papers is an issue, which means the heterogeneity in our review could have been due to different factors such as tools used to assess associated factors or the definitions used in their studies. For instance, between-study variability on the categorization of maternal age and BMI prevented us from pooling the effects of age and BMI on risk of developing HDP. In addition, for some of the associated factors there were not enough studies per associated factor to assess the association with HDP and not all SSA countries were represented in this review. Therefore, it need more comprehensive research to conduct in the region.

## Conclusions and recommendations

The results of the present research showed primiparity, having previous HDP, family history of HDP, and low maternal educational level are associated factors for developing HDP among women in SSA countries. Understanding those important associated factors in SSA women could help clinicians to identify pregnant women who are at increased risk of developing HDP. Even though several factors were found to have mixed, inconclusive or no association with HDP in our review, we strongly encourage other researchers to investigate these factors further.

In general, most of the associated factors could be identified through ANC visits and this could halt further complications to the mother as well as the foetus. Therefore, interventions need to be design to address these factors. The findings of this review can be used by SSA countries to develop a screening guideline or checklist for pregnant women during ANC visits. Therefore, revising the guidelines and service provision material can help health care providers on how to approach pregnant women and identify potential associated factors before serious complications occur. Moreover, it is essential to design effective interventions through multi-sector approaches that focus on primary prevention, mainly by screening high-risk women and improving women's literacy rate.

## Supporting information

**S1 Checklist. PRISMA checklist.**
(DOC)

**S1 Table. Searching terms for research articles.**
(DOCX)

**S2 Table. Quality assessment for research article.**
(DOCX)

**S1 File. Funnel plots for associated factors of HDP in SSA.**
(DOCX)

## Acknowledgments

The authors would like to thank the University of Newcastle, Australia, for providing us free digital access to the online library. We would also like to thank Mrs Debbie Booth for her assistance in designing the search strategy and help in searching databases.

## Author Contributions

**Conceptualization:** Maereg Wagnew Meazaw, Catherine Chojenta, Deborah Loxton.

**Data curation:** Maereg Wagnew Meazaw, Muluken Dessalegn Muluneh.

**Formal analysis:** Maereg Wagnew Meazaw.

**Investigation:** Maereg Wagnew Meazaw.

**Methodology:** Maereg Wagnew Meazaw, Catherine Chojenta, Muluken Dessalegn Muluneh, Deborah Loxton.

**Resources:** Maereg Wagnew Meazaw.

**Software:** Maereg Wagnew Meazaw.

**Supervision:** Catherine Chojenta, Deborah Loxton.

**Validation:** Maereg Wagnew Meazaw.

**Visualization:** Muluken Dessalegn Muluneh.

**Writing – original draft:** Maereg Wagnew Meazaw.

**Writing – review & editing:** Catherine Chojenta, Muluken Dessalegn Muluneh, Deborah Loxton.

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
