## [Decision Letter · Decision Letter 0]

26 Jun 2020

PONE-D-20-16762

Factors Associated with Hypertensive Disorders of Pregnancy in sub-Saharan Africa: A Systematic and Meta-analysis

PLOS ONE

Dear Dr. Meazaw,

Thank you for submitting your manuscript to PLOS ONE. After careful consideration, we feel that it has merit but does not fully meet PLOS ONE’s publication criteria as it currently stands. Therefore, we invite you to submit a revised version of the manuscript that addresses the points raised during the review process.

SPECIFIC ACADEMIC EDITOR COMMENTS: Three expert reviewers in the field handled your manuscript. We greatly thank them for their time and efforts. Although interest was found in your study, numerous major concerns arose during review. These comments relate to the need for correction of several wrong phrases or statements that must be supported by the literature; the novelty of this study is questionable whereby the problem of greater preeclampsia/eclampsia is known in developing countries - how does this review shed any additional insight in to why this occurs?; clarification is needed about the experimental design, presentation of the PRISMA results, and statistical analysis; and the English needs to be proofed and corrected by an expert in the language. Please address ALL comments in your revised manuscript.

We look forward to receiving your revised manuscript.

Kind regards,

Frank T. Spradley

Academic Editor

PLOS ONE

Journal Requirements:

2. We note you have included a table to which you do not refer in the text of your manuscript. Please ensure that you refer to Table 1 in your text; if accepted, production will need this reference to link the reader to the Table.

Reviewers' comments:

Reviewer's Responses to Questions

**Comments to the Author**

1. Is the manuscript technically sound, and do the data support the conclusions?

Reviewer #1: Partly

Reviewer #2: Yes

Reviewer #3: Partly

2. Has the statistical analysis been performed appropriately and rigorously? 

Reviewer #1: I Don't Know

Reviewer #2: Yes

Reviewer #3: Yes

3. Have the authors made all data underlying the findings in their manuscript fully available?

Reviewer #1: Yes

Reviewer #2: Yes

Reviewer #3: Yes

4. Is the manuscript presented in an intelligible fashion and written in standard English?

Reviewer #1: Yes

Reviewer #2: No

Reviewer #3: Yes

5. Review Comments to the Author

Reviewer #1: 1. 'Whereas, the 2014 International Society for the Study of Hypertension in Pregnancy (ISSHP) classified HDP as chronic hypertension, gestational hypertension, preeclampsia – de novo or superimposed on chronic hypertension and White coat hypertension (4)'. Recheck for clarity.

2. ''According to the World Health Organization (WHO) estimates, the incidence of preeclampsia in developing countries (2.8% of live births) is seven times higher than in developed countries (0.4% of live births) (6, 8). Eclampsia also increases the risk of maternal death both in developed (0.5-1.8%) and in developing countries (15%) (9).'' Having stated this as a problem, authors are expected to attempt to find answer to this in their own this review.

3. ''Similarly, a prospective cohort study of 763,795 primiparous women from the Swedish Medical Birth Register showed that the risk of developing preeclampsia was 14.7% in the second pregnancy for women who had had

preeclampsia in their first pregnancy.'' Authors should be bothered about this 'apples and oranges comparisons'' given the disparities in environmental. dietary and socioeconomic factors.

4. ''Although we cannot draw conclusions by doing a meta-analysis as the number of studies found per

variable was limited, our review highlights that doing physical activity during pregnancy'' recast doing physical activity

Reviewer #2: 1. I recommend English language editing of the manuscript to improve on the readability and flow.

2. Please refer to Table 1( Jennifer Murray et al) on page 8. Under risk factor(s) assessed, the third word should be "sprayed" and not "spayed".

3. Under the discussion on page 23 (line 16) there is an incomplete sentence: risk of maternal and foetal ???? in her first pregnancy....

4. In the PRISMA flow chart (Fig S1), I suggest that all articles identified in the electronic and hand searches should be included in the first box under "identification". Brief reasons should be given for the full text articles excluded (n=3).

Reviewer #3: Thank you for the opportunity to review this interesting manuscript. It is an important and interesting topic that can have the potential to improve outcomes for women in sub-Saharan Africa. There are however some comments I make for your consideration before I believe the paper is ready for publication.

Specific comments

1. The authors inconsistently and interchangeably used terms like ‘risk factors’, ‘associated factors’ and ‘determinant factors’. I would suggest to stick to the appropriate one, may be ‘associated factors’.

Introduction

2. At the end of page 3 and the beginning of page 4, you stated that ‘there are inconsistencies in reporting determinant factors in the literature…’ and ‘the studies conducted on risk factors for HDP across different SSA countries were at the health facility level and had small sample sizes.’ I would suggest substantiating these sentences with evidence. i.e. cite papers that reported risk factors inconsistently, conducted at facility level and had small sample size.

3. You can paraphrase the last sentence of the introduction section as ‘The aim of this study was to identify factors associated with HDP in SSA countries’

Methods

4. In your selection criteria, you were highly restrictive. You included only peer-reviewed papers. This could be one source of publication bias. You also excluded studies that were carried out specific to any types of HDP. However, in the abstract section you stated that “due to the limited number of studies found specific to each variable, there was inconclusive evidence for a relationship with a number of factors, such as nutrition and related factors…”. Could your exclusion criteria contribute to the limited number of studies?

5. You assessed publication bias by an asymmetry test. Since asymmetrical funnel plots can be caused by factors other than publication bias, I would suggest that you replace the term ‘publication bias’ with ‘small-study effects’ (see as an example the following: Sterne, J. A., et al. (2011). "Recommendations for examining and interpreting funnel plot asymmetry in meta-analyses of randomised controlled trials." British Medical Journal 343: d4002).

6. You stated that two reviewers independently screened, extracted, and assessed the quality of the articles. I would suggest you to report the level of agreement between the two reviewers perhaps using the Cohen’s Kappa (K) coefficient statistics.

Results

7. I would suggest you to report the number of studies assessed though each database separately. Now you have reported cumulative number.

8. As per PRISMA guidelines, I would suggest that you tell the reader the reason (s) for exclusion of studies at each stage in the flow diagram.

9. Your PRISMA flow chart is a bit confusing. Every paper supposed to pass through similar screening method. However, some of the papers for example hand searched papers included after duplication removed, and reference section reviewed papers included without eligibility checking according to the diagram. This would needs clarification or correction.

10. One of the major limitation of the manuscript is misinterpreting of the odds ratio. Throughout the results and discussions, you interpreted odds ratio as if it is risk ratio. For example, you interpreted odds ratio as “older were 5.3 times more likely to have HDP compared to younger women (AOR: 5.32; 95% CI: 2.55, 11.10)”; “younger women had a higher risk of developing HDP compared to older women (AOR: 17.0; 95% CI: 5.0, 73.2)”; and “ Primiparous women have nearly a twofold increased risk of developing HDP compared to multiparous women (OR: 1.78; 95% CI: 1.11, 2.44)”. You cannot interpret odds ration as these. See these papers ( Alexander Persoskie, Rebecca A.Ferrer: A Most Odd Ratio: Interpreting and Describing Odds Ratios American Journal of Preventive Medicine Volume 52, Issue 2, Pages 224-228; Ju Heon Kim, Min Young Kim, Soo Young Kim: In Hong Hwang, and En Jin Ka Misinterpreting Odds Ratio in the Articles Published in Korean Journal of Family Medicine. Korean J Fam Med. Volume 33, Issue 2, Pages 89–93.)

11. You reported that funnel plot and Egger’s test showed there was no publication bias in the included studies. I would suggest to include the funnel plot results in the result section so as the readers have the chance to evaluate it.

12. On page 14, in the last sentence, please change I2 to I2.

13. On page 18, please add MUAC to the VI sub-heading

Discussion

14. Some of your conclusions and recommendations were not supported by your result. You may need reconsider it.

6. PLOS authors have the option to publish the peer review history of their article (what does this mean?). If published, this will include your full peer review and any attached files.

Reviewer #1: **Yes: **Orish Ebere ORISAKWE

Reviewer #2: No

Reviewer #3: **Yes: **Fekede Asefa

---

## [Author Response · Author response to Decision Letter 0]

14 Jul 2020

Authors’ response to reviewers

1. Is the manuscript technically sound, and do the data support the conclusions?

Reviewer #`1: Partly

Response: For further clarification, the definition of HDP, the PRISMA flow diagram and the result section has been rewritten in revised version of manuscript.

Reviewer #1: 1. 'Whereas, the 2014 International Society for the Study of Hypertension in Pregnancy (ISSHP) classified HDP as chronic hypertension, gestational hypertension, and preeclampsia – de novo or superimposed on chronic hypertension and White coat hypertension (4)'. Recheck for clarity.

Response: Thank you for your feedback. We have now rewrite this sentence in this version of manuscript (line 49-56).

“Different international organizations and professional societies have different classifications of HDP (3-5). For instance, According to the American College of Obstetrics and Gynaecology (ACOG) classification , HDP are a group of diseases and conditions that occur during pregnancy, including preeclampsia and eclampsia, gestational hypertension, chronic hypertension, and preeclampsia superimposed on chronic hypertension (5). Whereas, according to the International Society for the Study of Hypertension in Pregnancy (ISSHP) classification, HDP classified as chronic hypertension, gestational hypertension, preeclampsia – de novo or superimposed on chronic hypertension and White coat hypertension (4).”

2. ''According to the World Health Organization (WHO) estimates, the incidence of preeclampsia in developing countries (2.8% of live births) is seven times higher than in developed countries (0.4% of live births) (6, 8). Eclampsia also increases the risk of maternal death both in developed (0.5-1.8%) and in developing countries (15%) (9).'' Having stated this as a problem, authors are expected to attempt to find answer to this in their own this review.

Response: Thank you for your feedback. It is true that our review did not address this problem. Therefore, we believe also this sentence is does not cover under our review and we omit this from the introduction section on the revised manuscript. 

3. ''Similarly, a prospective cohort study of 763,795 primiparous women from the Swedish Medical Birth Register showed that the risk of developing preeclampsia was 14.7% in the second pregnancy for women who had preeclampsia in their first pregnancy.'' Authors should be bothered about this 'apples and oranges comparisons'' given the disparities in environmental, dietary and socioeconomic factors.

Response: we accepted your feedback. We believed that other statement discussed the finding of our review and we omit this sentence in the revised version.

4. ''Although we cannot draw conclusions by doing a meta-analysis as the number of studies found per variable was limited, our review highlights that doing physical activity during pregnancy'' recast doing physical activity

Response: In our review, those studies that included used different classification for physical activity during pregnancy. For example, one study used the International Physical Activity Questionnaire (IPAQ) which designed and for adult age 15-69 years and categorized women into high, moderate and low groups. Whereas, the other defined physical activity as at least a conscious effort to stroll around participant’s home for not less than 20-30 minutes daily. The other two studies did not reported how physical activities defined in their studies. Therefore, as per your feedback we included this information in the final version of the manuscript as follow (line 449-454). 

“Although we cannot draw conclusions due to difference in categorization of level of physical activity during pregnancy and identified studies does not fulfil our meta-analysis criteria, this review highlights that doing physical activity during pregnancy was found to be protective against developing HDP. Regular physical activity is recommended for pregnant and postpartum women for maternal, foetal, and neonatal wellbeing (65-67). However, the relationship between HDP and the level of exercise needs further investigation.”

 Reviewer #2: 1. I recommend English language editing of the manuscript to improve on the readability and flow.

Response: Thank you for your constructive comments. We accepted the comment of language revision and the revised version of the manuscript proefread by University of Newcastle ELICOS teacher, Natalia Soeters. In addition, two of the co-authors are native English speaker and they revised for better flow and readability. 

2. Please refer to Table 1( Jennifer Murray et al) on page 8. Under risk factor(s) assessed, the third word should be "sprayed" and not "spayed".

Response: Thank you, we corrected in the revised version (Table 1). 

3. Under the discussion on page 23 (line 16) there is an incomplete sentence: risk of maternal and foetal ???? in her first pregnancy....

Response: The incomplete sentence has now completed and revised. We revised as follow (line 414-417).

“In fact, every woman will be at increased risk of complications in her first pregnancy, this review also support being primiparous found to be associated factor for developing HDP.”

4. In the PRISMA flow chart (Fig S1), I suggest that all articles identified in the electronic and hand searches should be included in the first box under "identification". Brief reasons should be given for the full text articles excluded (n=3).

Response: we have now revised and incorporated exclusion reasons in the PRISMA flow diagram (Page 9). 

Reviewer #3: Thank you for the opportunity to review this interesting manuscript. It is an important and interesting topic that can have the potential to improve outcomes for women in sub-Saharan Africa. There are however some comments I make for your consideration before I believe the paper is ready for publication.

Response: Thank you for your review and feedback. 

1. The authors inconsistently and interchangeably used terms like ‘risk factors’, ‘associated factors’ and ‘determinant factors’. I would suggest to stick to the appropriate one, may be ‘associated factors’.

Response: Thank you for your constructive feedback. As per your suggestion we used associated factors” consistently throughout this revised manuscript.

Introduction

2. At the end of page 3 and the beginning of page 4, you stated that ‘there are inconsistencies in reporting determinant factors in the literature…’ and ‘the studies conducted on risk factors for HDP across different SSA countries were at the health facility level and had small sample sizes.’ I would suggest substantiating these sentences with evidence. i.e. cite papers that reported risk factors inconsistently, conducted at facility level and had small sample size.

Response: We have now included reference for the above-mentioned sentences (line 74& 77). 

3. You can paraphrase the last sentence of the introduction section, as ‘The aim of this study was to identify factors associated with HDP in SSA countries’

Response: We have now revised the last sentence as per your direction (line 79-80). 

4. In your selection criteria, you were highly restrictive. You included only peer-reviewed papers. This could be one source of publication bias. You also excluded studies that were carried out specific to any types of HDP. However, in the abstract section you stated that “due to the limited number of studies found specific to each variable, there was inconclusive evidence for a relationship with a number of factors, such as nutrition and related factors…”. Could your exclusion criteria contribute to the limited number of studies. 

Response: Thank you for the feedback. It is true it may contribute if we follow restrictive procedure. However, besides peer reviewed research we tried hand search and reference searching to get additional research; moreover, we have tried an attempt to check different libraries if there is any research conducted on this area. However, we could not find any more studies that related to the aim of our study. In addition, our review aim to identify the associated factors for HDP as general. That is why we included studies only conducted on HDP. 

5. You assessed publication bias by an asymmetry test. Since asymmetrical funnel plots can be caused by factors other than publication bias, I would suggest that you replace the term ‘publication bias’ with ‘small-study effects’ (see as an example the following: Sterne, J. A., et al. (2011). "Recommendations for examining and interpreting funnel plot asymmetry in meta-analyses of randomised controlled trials British Medical Journal 343: d4002).

Response: Thank you for your feedback and provide us a reference for further reading. We accepted and corrected in the revised version. 

6. You stated that two reviewers independently screened, extracted, and assessed the quality of the articles. I would suggest you to report the level of agreement between the two reviewers perhaps using the Cohen’s Kappa (K) coefficient statistics.

Response: The two independent reviewer were used common agreed extraction criteria and at the end, we checked the difference and reach to consensus by discussion. Unfortunately, we did not use any software application to measure the level of agreement. 

Results

7. I would suggest you to report the number of studies assessed though each database separately. Now you have reported cumulative number.

Response: We added the number of studies for each databases (page 8).

8. As per PRISMA guidelines, I would suggest that you tell the reader the reason (s) for exclusion of studies at each stage in the flow diagram.

Response: We included the reason for exclusion in revised version (page 8).

9. Your PRISMA flow chart is a bit confusing. Every paper supposed to pass through similar screening method. However, some of the papers for example hand searched papers included after duplication removed, and reference section reviewed papers included without eligibility checking according to the diagram. This would needs clarification or correction.

Response: Thank you and we have now corrected the PRISMA flow diagram in this revised version. 

10. One of the major limitation of the manuscript is misinterpreting of the odds ratio. Throughout the results and discussions, you interpreted odds ratio as if it is risk ratio. For example, you interpreted odds ratio as “older were 5.3 times more likely to have HDP compared to younger women (AOR: 5.32; 95% CI: 2.55, 11.10)”; “younger women had a higher risk of developing HDP compared to older women (AOR: 17.0; 95% CI: 5.0, 73.2)”; and “ Primiparous women have nearly a twofold increased risk of developing HDP compared to multiparous women (OR: 1.78; 95% CI: 1.11, 2.44)”. You cannot interpret odds ration as these. See these papers ( Alexander Persoskie, Rebecca A.Ferrer: A Most Odd Ratio: Interpreting and Describing Odds Ratios American Journal of Preventive Medicine Volume 52, Issue 2, Pages 224-228; Ju Heon Kim, Min Young Kim, Soo Young Kim: In Hong Hwang, and En Jin Ka Misinterpreting Odds Ratio in the Articles Published in Korean Journal of Family Medicine. Korean J Fam Med. Volume 33, Issue 2, Pages 89–93.)

Response: We accepted your feedback and revised in this version of manuscript. 

11. You reported that funnel plot and Egger’s test showed there was no publication bias in the included studies. I would suggest to include the funnel plot results in the result section so as the readers have the chance to evaluate it.

Response: Thank you and we found that is important and we added the funnel plot as supplementary document.

Supplementary 

12. On page 14, in the last sentence, please change I2 to I2.

Response: we corrected in the revised

13. On page 18, please add MUAC to the VI sub-heading

Response: we have added MUAC on the sub-heading (line 266).

Discussion

14. Some of your conclusions and recommendations were not supported by your result. You may need reconsider it.

Response: Thank you for your input. We revised the conclusions and recommendations section to be in line with our finding.

---

## [Decision Letter · Decision Letter 1]

28 Jul 2020

Factors Associated with Hypertensive Disorders of Pregnancy in sub-Saharan Africa: A Systematic and Meta-analysis

PONE-D-20-16762R1

Dear Dr. Meazaw,

We’re pleased to inform you that your manuscript has been judged scientifically suitable for publication and will be formally accepted for publication once it meets all outstanding technical requirements.

Kind regards,

Frank T. Spradley

Academic Editor

PLOS ONE

Reviewers' comments:

Reviewer's Responses to Questions

**Comments to the Author**

1. If the authors have adequately addressed your comments raised in a previous round of review and you feel that this manuscript is now acceptable for publication, you may indicate that here to bypass the “Comments to the Author” section, enter your conflict of interest statement in the “Confidential to Editor” section, and submit your "Accept" recommendation.

Reviewer #1: All comments have been addressed

Reviewer #2: All comments have been addressed

Reviewer #3: All comments have been addressed

2. Is the manuscript technically sound, and do the data support the conclusions?

Reviewer #1: Yes

Reviewer #2: Yes

Reviewer #3: Yes

3. Has the statistical analysis been performed appropriately and rigorously? 

Reviewer #1: Yes

Reviewer #2: Yes

Reviewer #3: Yes

4. Have the authors made all data underlying the findings in their manuscript fully available?

Reviewer #1: Yes

Reviewer #2: Yes

Reviewer #3: Yes

5. Is the manuscript presented in an intelligible fashion and written in standard English?

Reviewer #1: Yes

Reviewer #2: Yes

Reviewer #3: Yes

6. Review Comments to the Author

Reviewer #1: The quality of this manuscript has improved. Authors have adequately and satisfactorily responded to the queries and concerns raised earlier.

Reviewer #2: (No Response)

Reviewer #3: I would like to thank the authors for addressing almost all of my comments. However, there is still a problem in the interpretation of the odds ratio. The authors still using the terms ' increase risk' and 'more likely' in multiple places while their measures of association were odds ratio. These terms are not the language of odds ratio. When you use these terms to interpret odds ratio, you are wrongly overestimating risk of HDP. I would still suggest you to critically look at the interpretations of your odds ratio.

Kind regards

7. PLOS authors have the option to publish the peer review history of their article (what does this mean?). If published, this will include your full peer review and any attached files.

Reviewer #1: **Yes: **Orish Ebere ORISAKWE

Reviewer #2: No

Reviewer #3: **Yes: **Fekede Asefa

---

## [Editor Report · Acceptance letter]

30 Jul 2020

PONE-D-20-16762R1 

Factors Associated with Hypertensive Disorders of Pregnancy in sub-Saharan Africa: A Systematic and Meta-analysis 

Dear Dr. Meazaw:

I'm pleased to inform you that your manuscript has been deemed suitable for publication in PLOS ONE. Congratulations! Your manuscript is now with our production department. 

Kind regards, 

on behalf of

Dr. Frank T. Spradley 

Academic Editor

PLOS ONE